# A Novel Bacillus Velezensis for Efficient Degradation of Zearalenone

**DOI:** 10.3390/foods13040530

**Published:** 2024-02-09

**Authors:** Yijia Li, Songbiao Chen, Zuhua Yu, Jie Yao, Yanyan Jia, Chengshui Liao, Jian Chen, Ying Wei, Rongxian Guo, Lei He, Ke Ding

**Affiliations:** 1Luoyang Key Laboratory of Live Carrier Biomaterial and Animal Disease Prevention and Control, College of Animal Science and Technology, Henan University of Science and Technology, Luoyang 471023, China; 13462205724@163.com (Y.L.); chensongbiao@126.com (S.C.); yzhd05@163.com (Z.Y.); yj18403798226@163.com (J.Y.); jiayanyan0120@163.com (Y.J.); liaochengshui33@163.com (C.L.); chenillejian@163.com (J.C.); 9906127@haust.edu.cn (Y.W.); guorongxian520@163.com (R.G.); helei4280546@163.com (L.H.); 2Laboratory of Functional Microbiology and Animal Health, College of Animal Science and Technology, Henan University of Science and Technology, Luoyang 471003, China; 3Ministry of Education Key Laboratory for Animal Pathogens and Biosafety, Zhengzhou 450000, China

**Keywords:** zearalenone, *Bacillus velezensis*, detoxification, whole-genome sequence

## Abstract

Zearalenone (ZEN) is considered one of the most serious mycotoxins contaminating grains and their by-products, causing significant economic losses in the feed and food industries. Biodegradation pathways are currently considered the most efficient solution to remove ZEN contamination from foods. However, low degradation rates and vulnerability to environmental impacts limit the application of biodegradation pathways. Therefore, the main research objective of this article was to screen strains that can efficiently degrade ZEN and survive under harsh conditions. This study successfully isolated a new strain L9 which can efficiently degrade ZEN from 108 food ingredients. The results of sequence alignment showed that L9 is *Bacillus velezensis*. Meanwhile, we found that the L9 degradation rate reached 91.14% at 24 h and confirmed that the primary degradation mechanism of this strain is biodegradation. The strain exhibits resistance to high temperature, acid, and 0.3% bile salts. The results of whole-genome sequencing analysis showed that, it is possible that the strain encodes the key enzyme, such as chitinase, carboxylesterases, and lactone hydrolase, that work together to degrade ZEN. In addition, 227 unique genes in this strain are primarily involved in its replication, recombination, repair, and protective mechanisms. In summary, we successfully excavated a ZEN-degrading, genetically distinct strain of *Bacillus velezensis* that provides a solid foundation for the detoxification of feed and food contamination in the natural environment.

## 1. Introduction

Mycotoxin contamination in food and feed has become a serious global problem over the past few decades [1]. As one of the four major mycotoxins, zearalenone (ZEN) not only causes widespread contamination of grains such as wheat, barley, and soybeans, but is often present in animal-origin foods such as meat, eggs, and dairy products in the form of ZEN and its related derivatives [2]. At the same time, it is also found in common food sources such as cooking oil, corn protein powder, and other foods [3]. ZEN is structurally similar to estrogen [4] and affects the endocrine system in humans [5] and animals [6] by competing with the estrogen receptor (ER). Reproductive toxicity, immunotoxicity [7], and liver and kidney toxicity [8] can occur when food contaminated with ZEN and its associated derivatives are ingested. ZEN has been reported to cause cardiovascular aging-related injuries [9]. According to data from the Food and Agriculture Organization and the World Health Organization, mycotoxin contamination rates are as high as 25% in food crops worldwide [10]. Analysis of ZEN in about half a million grains and nuts in large studies by the European Food Safety Agency and around the world showed actual contamination rates between 60% and 80% [10]. ZEN pollution levels are well above the EU and Codex Alimentarius Commission limits [11]. Then, mycotoxins were studied in our feed samples between 2017 and 2021. The study found that ZEN occurs quite frequently in temperate regions of the northeast of China, with a positive test rate of 55.67% [12]. It is clear that ZEN pollution has worsened year after year over time. Therefore, an in-depth study of ZEN detoxification techniques is essential to ensure food safety and improve the quality of feed production.

With regard to the degradation technology of ZEN, it mainly consists of three methods: physical [13], chemical [14], and biological [15]. Biological detoxification, as an environmentally friendly and low-cost method, shows obvious superiority and great application potential in eliminating the ZEN mycotoxin. Zhou et al. found that coding genes Oxa (a ZEN-degradation enzyme isolated from SM04) were cloned into *Lactobacillus acidophilus* ATCC4356 and expressed with a degradation rate of 42.95% at 12 h (ZEN: 20 μg/mL) [16]. Phosphoryl transferase [17] in *Bacillus subtilis* Y816 and Cota laccase [18] in *Bacillus subtilis* both have the ability to reduce ZEN toxicity. Although there are many microbial species and enzymes that can eliminate ZEN, they are relatively weak at degrading ZEN, mainly because these microbes are particularly sensitive to the environment. Therefore, it is of great practical significance to select strains with high degradation efficiency, safety, and effectiveness, and with a clear genetic background that can degrade ZEN.

In this study, we screened for microorganisms that could degrade ZEN from food materials collected in China, and isolated a strain with efficient ZEN degradation, which was identified as *Bacillus velezensis* L9 based on 16S rRNA analysis. Furthermore, the key mechanisms for the breakdown of ZEN were explored, and the resistance of the strain to high-temperature, acidic, and choline salt conditions was investigated. Genome-wide sequencing analysis of L9 strains revealed the complexity of their genomes and identified the genetic background for ZEN degradation. In combination with detailed bioinformatics analysis, gene clusters encoding CAZymes and secondary metabolites in L9 strains were identified, and enzymes capable of degrading ZEN were unearthed.

## 2. Materials and Methods

### 2.1. Isolation of Zearalenone-Degrading Strains

A sample of 108 food ingredients was collected from a school canteen in Luoyang, Henan province. The samples (5 g) were placed in a 200 mL cone jar, 100 mL of sterile water was added, then they were stirred for 1 h at 37 °C and left to rest for 30 min. The suspension was mixed with 900 μL of sterile saline and then diluted with gradient dilution. A suitable concentration of diluent was extracted and 100 μL was uniformly coated on LB (lysogeny broth) agar medium containing ZEN (10 μg/mL) and incubated at 37 °C for 24 h. A colony was selected and cultured multiple times on the LB agar plate until the strain was purified. The purified single colony was inoculated into LB liquid medium (containing 10 μg/mL ZEN). An LB liquid medium containing the same amount of ZEN was used as a control. The residue of ZEN was determined by HPLC by placing it in a rocker oscillation culture at 37 °C and 200 rpm for 48 h. After detection, the strain with the highest degradation rate was selected as the target strain, to identify and explore its biological characteristics.

HPLC system: Waters XBridge C18 (5 µm 4.6 × 250 mm) column. Mobile phase: water: methanol (20:80); velocity: 0.8 mL/min; column temperature: 35 °C; sample entry: 10 µL. The fluorescence detector excitation wavelength was 274 nm, and the emission wavelength was 440 nm [19].

ZEN degradation rate = (ZEN content in the blank control group − ZEN content in sample group)/ZEN content in the blank control group × 100%.

### 2.2. 16S rDNA Sequencing of L9 Strain

Based on the instructions in the bacterial genome kit, the genome of the strain was extracted, and PCR fragments were amplified using bacterial generic primers 27F (5′-AGTTGTGTGTGCTCAG-3′) and 1492R (5′-GGTGTGTGCTCAG-3′), modeled on the extracted DNA. PCR reaction conditions: initial denaturation at 94 °C for 4 min, followed by 35 cycles consisting of 94 °C denatured 30 s, 55 °C annealing for 30 s, and 72 °C extension for 1 min; end extension at 72 °C for 10 min; 4 °C save. The amplified PCR products were sent to Shanghai Bioengineering Co. The sequencing results were compared to BLAST on NCBI, higher homologous sequences were selected, and the system evolution tree was constructed using Tree Visualization By One Table (TVBOT: https://www.chiplot.online/tvbot.html accessed on 25 February 2023) [20].

### 2.3. Morphological Identification of L9 Strain

The strain was inoculated on LB solid medium and incubated at 37 °C for 24 h. Then the colony morphology was observed, and samples from a single colony were selected for Gram staining to observe the micromorphology.

### 2.4. The Adsorption Ability of L9 Strain to ZEN

The strain L9 was inoculated into LB liquid medium and then shaken for 24 h at 37 °C and 200 rpm. The bacterial solution was divided into two parts, with one stored in a refrigerator at 4 °C and the other in high-temperature sterilization at 121 °C for 30 min. The two groups of bacteria were each treated with 2 mL LB liquid medium of the culture, centrifuged at 12,000 rpm, then washed three times with sterile PBS buffer, re-suspended with 2 mL of sterile PBS buffer, ZEN was added with an endpoint concentration of 10 μg/mL, and 2 mL of sterile ZEN with an endpoint concentration of 10 μg/mL was added as a control; then, samples were oscillated at 200 rpm for 24 h per sample. After the culturing process was completed, the ZEN residues in the medium were measured by HPLC.

### 2.5. Validation of Degradation Mechanisms of L9 Strain

Strain L9 was cultured, and a stable fermentation solution was obtained. The bacteria and supernatant were separated by centrifugation at a speed of 12,000 rpm for 10 min using a high-speed centrifuge. The strain was then treated with the following samples: (1) a volume of 2 mL of supernatant was taken and stored at 4 °C; (2) treated with 2 mL of supernatant plus protease K (65 °C, water bath for 2 h); (3) a volume of 2 mL of supernatant was added to protease K and 1% SDS (65 °C, water bath for 2 h); (4) heated 2 mL of supernatant (100 °C, water bath for 2 h); (5) the precipitated bacteria were rinsed 3 times with PBS buffer and then suspended in PBS using a cellular sonographer crusher at 12,000 rpm centrifugation for 10 min as a cell pellet.

The six samples were added to ZEN with a final concentration of 10 µg/mL and 2 mL of aseptic PBS of ZEN with a final concentration of 10 µg/mL was used as a blank control and oscillated for 24 h at 37 °C and 200 rpm, with 3 repetitions for each sample. After the culture process was completed, the ZEN residues in the medium were measured by HPLC.

### 2.6. Temperature, Acid, and Bile Salt Tolerance Assay of L9 Strain

#### 2.6.1. Determination of Strain Growth Curve

In 100 mL of LB liquid medium, 2% fermentation solution was used for inoculation and then incubated at 37 °C for 24 h and tested every 2 h. The growth curve is drawn with time as the transverse coordinate and absorbance as the longitudinal coordinate.

#### 2.6.2. Effects of Temperature on L9 Strain

A volume of 1 mL of the L9 strain was placed in a 1.5 mL centrifuge tube in a water bath at 50 °C, 60 °C, 70 °C, and 80 °C and was tested after 3 h. A water bath at 37 °C was used as the blank control.

#### 2.6.3. Effects of pH on L9 Strain

The L9 strain was treated with 100 mL of liquid LB medium with pH values of 2.0, 3.0, 4.0, 5.0, and 6.0 at 2% inoculation and incubated at 37 °C after 3 h. LB liquid medium with a pH of 7.0 was used as the blank control.

#### 2.6.4. Effects of Bile Salts on L9 Strain

The L9 strain was inoculated with 0.1%, 0.2%, and 0.3% bovine bile salts in 100 mL of LB liquid medium at 37 °C and tested after 3 h. LB liquid medium without bile salts was used as the blank control.

In the above experiments, spectroscopic measurements were performed at a wavelength of 600 nm using an enzyme marker, and live bacteria counts were performed.

### 2.7. Genome Sequencing of L9 Strain

The strain L9 was inoculated into LB liquid medium and centrifuged 24 h after culturing to collect bacteria. Bacterial DNA extraction kits were then used to extract DNA, which was sent to Bioengineering Ltd. (Shanghai, China) for whole-genome sequencing.

### 2.8. Quality Control and Assembly of Genome Sequencing Data of L9 Strain

The data generated by offline sequencing are two-end, and the sequencing sequence contains barcode sequences, as well as primers and connector sequences added to the sequencing process. First, the primer connector sequence needs to be removed, then the paired reads (merges) are stitched into a sequence based on the overlap relationship between the two- end sequences, and then the samples are identified and differentiated according to the barcode label sequence to obtain data for each sample [21]. To ensure the accuracy of subsequent analyses, we needed to evaluate and process the collected data in depth as an initial stage of the experiment. The raw data were screened using FastQC 0.11.2 [22] to exclude sequenced joints and primer sequences in reads, as well as reads with an average mass of less than Q5 and reads with N numbers greater than 5. Finally, spades [23] were used to combine second-generation sequencing data. Contig complement GAP was repaired by using GapFiller 1.11 [24], and sequence corrections were made using Pilon technology to correct editing errors and missing small segment insertions that may have occurred during the splicing process. The experimental data are available on NCBI (Accession GCF_033022835.1).

### 2.9. Gene Component Prediction of L9 Strain

The NCBI-PGAP prokaryotic genomic annotation pipeline was employed in this study to anticipate assembly outcomes based on genetic constituents. Data on tRNA and rRNA sequencing were acquired from the prokka 1.10 genome [25]. NCBI has developed PGAP, an automated prokaryotic genomic annotation pipeline that merges a gene prediction algorithm from the ground up with a homology-based approach. Through the utilization of the protein family model for structural and functional annotation, this compilation of evidence encompasses the HMM Model, BLAST-based protein family (Blast Rules), and conservative domain database architecture (CDD) in a hierarchical manner. The genome sequence of individual samples was comprehensively characterized by combining genome sequencing, coding gene prediction, and non-coding RNA into GBK files, utilizing the CGView 1.0 [26].

### 2.10. Basic Genome Annotation of L9 Strain

The anticipated gene sequences were cross-referenced with the NR, GO, COG, and KEGG databases through the utilization of the BLAST 2.2.28, followed by a comparison of the predicted gene amino acid sequences with the Pfam database using the HMMER 3.1b1 to acquire annotated information regarding the predicted genes. The primary objective of annotating the function of the protein-coding gene was to conduct a functional analysis of all protein-coding genes, thereby examining the strain at the molecular level.

The KEGG (Kyoto Encyclopedia of Genes and Genomes) database [27] is a comprehensive repository of biological systems that amalgamates genomic, chemical, and systemic functional data, and following the annotation of genes, categorizes KEGG metabolic pathways according to their correlation with the pathway. The database is connected to the following URL: https://www.genome.jp/kegg (accessed on 25 February 2023).

COG [28] (Clusters of Orthologous Groups of Proteins) is the NCBI’s gene-based lineage homologous relationship annotation system, and COG focuses on prokaryotes. COG combines evolutionary relationships to divide homologous genes from different species into different ortholog clusters. Genes from the same orthologs have the same function, so that functional annotations can be inherited directly from other members of the same COG cluster. Database link: https://www.ncbi.nlm.nih.gov/COG (accessed on 26 February 2023).

The GO (Gene Ontology) database provides standardized descriptions of genetic products, including molecular function, biological process, and cellular component, which are simply annotated to provide a snapshot of the biological functions, pathways, or cell localization of differential gene enrichment through GO enrichment analysis. The database can be accessed through this link: https://www.geneontology.org (accessed on 27 February 2023).

Comparison with the NR library allowed us to examine the approximation of species’ transcriptional sequences to similar species, as well as the functional information of homologous sequences.

### 2.11. Prediction of Genes Encoding for CAZymes and Secondary Metabolites

The predicted protein sequences of *Bacillus velezensis* L9 were aligned to the Carbohydrate-Active Enzymes Database (CAZy) [29] using OmicsBox 2.0.10. Clusters of secondary metabolite genes were identified with the aid of antiSMASH 6.0.1 [30].

### 2.12. Average Nucleotide Consistency (ANI) Analysis

The full name of ANI is average nucleotide identity. Average nucleotide identity is mainly used to assess the relatedness between species at the genome-wide level [31]. ANI analysis is simple, fast, and accurate. Based on the results of 16S rRNA gene sequence comparison, the strains with the highest 16S rRNA gene sequence similarity to strain L9 were selected, and the genome sequences of these 20 strains were downloaded from the assembly database of NCBI, and the JSpeciesWS [32] (http://jspecies.ribohost.com/jspeciesws/ accessed on 16 March 2023) was used to calculate the ANI values between each strain and strain L9. The ANI threshold for species delimitation is generally considered to be 95%, above which species are the same.

### 2.13. Pan-Genome Analysis

We compared genomic sequences of eight proximal *Bacillus velezensis* using the Mugsy 1.2.3, and performed an in-depth genome-wide and core genome analysis using the automated software for prokaryotic pan-genomics analysis (PGAP) [33]. Genomes were classified using the gene family (GF) method and these data were imported into the PanGP (http://PanGP.big.ac.cn accessed on 19 March 2023) [34]. Based on Heap’s law and the exponential law, we fitted the pan-genome and core genome signature curves, respectively. Based on the distribution of gene sets in the genomes of eight strains, we mapped the Pan-genome petals to show clustering between strains. Next, we used the TreeBeST 1.9. 2 software and maximum likelihood to compute the core/pan analysis results and set bootstrap parameters to build a systematic tree 1000 times [35].

## 3. Results

### 3.1. Isolation and Identification of L9 Strain

Using ZEN as the sole carbon source, 14 strains with the potential to degrade ZEN were screened from 108 food sources. The 14 strains were inoculated (10 μg/mL) in liquid medium containing ZEN and their ZEN toxins were detected by HPLC. As can be seen from Figure 1, the L9 strain isolated from moldy corn had the highest degradation rate of 91.14%, while the remaining 13 strains all had degradation rates of less than 56.8%.

As can be seen from Figure 2A, the sequence of the L9 strain was compared with the NCBI, revealing that the L9 strain is highly homologous to *Bacillus velezensis*. A phylogenetic tree was constructed using the TVBOT by selecting strains with higher sequence homology. Further identification of *Bacillus velezensis* L9 was performed using an optical microscope. The morphology of strain L9 (Figure 2B,D) was large colony, milky white color, rough wrinkle of colony surface, opaque colony, and irregular edge. Observation by optical microscope (Figure 2C,E) showed that the morphology of *Bacillus velezensis* L9 was short rod with blunt ends and oval spore with no expansion of spore sac. Therefore, the L9 strain was identified as *Bacillus velezensis*.

### 3.2. Analysis of Mechanism of Degradation of ZEN by L9 Strain

As can be observed from Figure 3, living cell bacteria exhibit a greater adsorption capacity than autoclaved bacteria. The degradation rate of the L9 supernatant was significantly higher than that of the L9 cell pellet, suggesting that the ability of the strain to degrade ZEN occurred mainly in the cell pellet. Additionally, the active substance of the strain to degrade ZEN occurred mainly outside the cell. The efficiency of the L9 supernatant degradation of ZEN was reduced to 47.61% after treatment with protease K. After co-treatment with protease K and 1% SDS, the degradation efficiency of ZEN decreased to 20.16%. The degradation efficiency of the strain decreased to 14.83% after a 100 °C bath for 2 h. The results showed that both *Bacillus velezensis* L9 and its supernatant showed detoxification of ZEN, suggesting that *Bacillus velezensis* L9 detoxification of ZEN includes both adsorption and degradation functions.

### 3.3. Temperature, Acid, and Bile Salt Tolerance of L9 Strain

The growth curve of strain L9 (Figure 4A) shows that at 0–4 h the growth rate of the strain was slow, at 4–12 h the growth rate was fast and after 12 h the growth rate of the strain was stable. The growth pattern of this strain shows that when it enters the gut, it can be activated quickly and grow and multiply in the gut. By studying the effects of L9 strains at different temperatures, pH, and bile salt concentrations, we were able to assess their biological properties. As can be seen from Figure 4B, the L9 strain still exceeded 3.78 × 10^7^ CFU/mL despite a gradual decrease in the number of live bacteria as the temperature increased. This shows that the L9 strain was highly resistant to high temperatures. According to the pH data, when the pH of the starter was 2, the number of live bacteria decreased significantly after 3 h of fermentation. Nevertheless, the number of live bacteria exceeded 2.43 × 10^7^ CFU/mL. The discovery suggests that the L9 strain has excellent acid-resistance properties, meaning it can survive well in the gut of both humans and animals. Experimental data on bile salt resistance showed that the L9 strain remained viable at 4.1 × 10^7^ CFU/mL or higher after 3 h of culture in 0.1%, 0.2%, and 0.3% bile salt environments. The results show that the L9 strain was able to tolerate 0.3% bile salts.

### 3.4. Genomic Overview of L9 Strain

The assembly results (Table 1) indicate that the genome of *Bacillus velezensis* L9 forms a closed DNA loop, measuring 3,974,499 bp and containing 46.25% GC. The prediction encompassed 4082 protein-coding genes, comprising tRNA, rRNA, sRNA, and transfer ribonucleic acid, responsible for the transportation of amino acids. The average length of the predicted genes was 872.52 bp, with the shortest genes at 56 bp and the longest genes at 16,302 bp. Figure 5 illustrates the mapping of chromosome loops, illustrating the distribution of components like genes, annotation information, GC content, etc., across the entire *Bacillus velezensis* L9 genome.

### 3.5. Genomic Basis Functional Annotation Results

By comparing the genes with databases to ascertain the gene function, we can examine the molecular-level biological attributes and functionalities of *Bacillus velezensis* L9, thereby establishing a theoretical foundation for the strain’s development and utilization.

#### 3.5.1. COG Database Annotation Results

COG is the NCBI’s homologous relationship annotation system based on genes, which can be classified into 25 categories by function. The function of the strain can be inferred by comparing protein sequences to the corresponding COG functional categories. Comparing the CDS sequence of *Bacillus velezensis* L9 to the COG database (Figure 6A), 2927 gene annotations were found. Among all the COG annotated genes, general function prediction had the highest gene count of 322, representing 11.01% of the total. This was followed by 279 genes with function unknown, which accounted for 9.53% of the annotated genes, suggesting that there is a lot of room for exploration of the strain. Furthermore, there were 274 genes involved in amino acid transport and metabolism, accounting for 9.36% of the annotated genes, while transcription comprised 252 genes, representing 8.61% of the annotated genes. Carbohydrate transport and metabolism included 196 genes, or 6.70% of the annotated genes.

#### 3.5.2. GO Database Annotation Results

The GO annotation aims to capture standardized functional description information that can help us better understand the biological implications behind genes. Figure 6B shows the annotated results of the protein-coding genes in the L9 genome of *Bacillus velezensis* in the GO database. A total of 3408 genes were annotated by the GO database: 2102 genes associated with biological processes, 418 genes associated with cellular response, and 888 genes associated with molecular function. In addition, they can be further classified into 34 secondary functional categories in the GO database. Of the 19 GO level II functional classes included in biological processes, the two categories with the highest number of protein-coding genes were cell process and metabolic process, with 670 and 580, respectively. Of the five GO secondary functional categories included in cellular components, the largest number of protein-coding genes was found in the membrane functional category, at 225. Only a handful of protein-coding genes are labeled in the functional classifications of protein-containing complexes and organelles. Molecular function contains the highest number of protein-coding genes in the 10 GO secondary functional categories, with 496 and 223, respectively, in the catalytic activity and binding function categories.

#### 3.5.3. KEGG Database Annotation Results

KEGG (Kyoto Encyclopedia of Genes and Genomes) is a relatively complete database of biological systems that integrates genomic, chemical, and systemic functional information. KEGG metabolic pathway annotation makes it easier to identify genes associated with a particular type of function. The results of the KEGG database annotation of protein-coding genes in the L9 genome of *Bacillus velezensis* are shown in Figure 6C, where the horizontal coordinates represent the level 2 classification of KEGG pathways, and the longitudinal coordinates represent the number of genes under annotation of the classification. There are 1937 protein-coding genes enriched in the genome of the L9 strain, that can be split into five major metabolic of pathways: cellular processes, environmental information processes, genetic information processes, metabolism, and organic systems. These can be further subdivided into 31 KEGG secondary functional categories. The genes most involved in metabolism were concentrated in carbohydrate metabolism, overview, and amino acid metabolism pathways, with 225, 215, and 209 gene annotation results, respectively. This was followed by the environmental information processing class pathway, which was predominantly enriched in the membrane transport and signal transport pathways, with 157 and 127 annotated results, respectively. Thus, strain L9 has a rich metabolic pathway and is very active in metabolism.

#### 3.5.4. NR Database Annotation Results

NR (NCBI non-redundant protein sequences) is the NCBI’s official collection of non-redundant protein sequences, which includes translation sequences of all non-redundant GenBank CDS, PDB (Protein Databank) protein databases, Swissport protein databases, and protein sequences from databases such as PIR (Protein Information Resource) and PRF (Protein Research Foundation). By comparing with the NR library, we were able to analyze the similarity between species’ transcriptional sequences and similar species, along with the functional details of homologous sequences. Comparing strain L9 with the NR database, strain L9 is highly homologous to *Bacillus velezensis*, as shown in Figure 6D.

### 3.6. CAZy Classification Result

The primary objective of CAZy (Carbohydrate-Active Enzymes Database) is to investigate and categorize enzymes that exhibit carbohydrate activity. The database offers a wide range of enzymes related to the breakdown, alteration, and manufacture of glycoside bonds. As shown in Figure 6E, a total of 80 gene-coding-protein domains in the strain L9 genome belong to the CAZy family, including glycoside hydrolases (GHs, 28), glycosyl transferases (GTs, 24), carbohydrates (CEs, 19), auxiliary activities (AAs, 6), and polysaccharides (PLs, 3). There were more GHs and GTs, accounting for about 35% and 30% of the gene family, respectively. In this family of genes, both GH18 and GH23 contain genes that encode chitinase. GH1 and other enzymes encode the degradation of cellulose and hemicellulose. The breakdown of xylitol is attributed to the genes present in GH1 and GH51, while enzymes encoding peptidoglycan degradation involve genes GH18, GH23, and CE4. GH1, GH16, GH26, and GH51 all contain encoding of enzymes involved in the breakdown of glucan. Drawing from this, we deduce that strain L9 possesses the capacity to decompose a diverse array of compounds such as chitin, cellulose, hemicellulose, xylitol, peptidoglycan, and glucan.

### 3.7. Secondary Metabolite Analysis

The secondary metabolites of *Bacillus velezensis* have a strong correlation with its antimicrobial activity. In this study, the genome of strain L9 was analyzed using antiSMASH 6.0, which encodes 15 gene clusters for the synthesis of secondary metabolites, as shown in Figure 6F. Out of these clusters, seven successfully identified clusters were either fully or very similar, whereas the remaining eight clusters failed to match any of the identified clusters. Table 2 shows the presence of four substances showing antifungal effects in these secondary metabolites: surfactin, fengycin, bacillibactin, and bacilysin. In addition, strain L9 can synthesize secondary metabolites that inhibit bacterial growth, including polyketones such as bacillaene, macrolactin H, and difficidin. Strain L9 may also encode five secondary metabolites with low or unknown similarity, of which three polyketone compounds were 38%, 20%, and 7% similar to plipastatin, fengycin, and butirosin A/butirosin B, respectively. The database has not yet matched clusters of similar compounds for two terpenes, one T3PKS, one ladderane, and one NRPS, which could mean they are unknown compounds. Strain L9 encodes a gene pool of a variety of antibiotics, suggesting that the strain may use these antimicrobial agents to achieve its antimicrobial properties.

### 3.8. Average Nucleotide Identity (ANI) Analysis

Based on the results of the whole-genome sequence comparison, 20 strains with the highest similarity to strain L9 were selected, and the genome sequences of these 20 strains were downloaded from the Assembly database of NCBI, and then the ANI values between strain L9 and the selected 20 strains were calculated using the JSpeciesWS. The results showed that the ANI values of strain L9 and *Bacillus velezensis* GH1-13 were higher than 99% (99.02%), followed by *Bacillus velezensis* M75, which was 98.70%. This indicates that strain L9 may belong to the same species as *Bacillus velezensis* GH1-13 (Figure 7). According to the ANI analysis, the L9 strain was identified as *Bacillus velezensis* in terms of the similarity of the genomes calculated at the genome level.

### 3.9. Pan-Genome Analysis

Genomic information and gene sequences of seven closely related strains were extracted from the NCBI RefSeq database, and core genes and specific genes between L9 and the seven closely related strains (DTU001, YA215, L-S60, YC-89, FC02, M75, GH1-13) were obtained using Roray technology. As shown in Figure 8A, the pan-genomic modeling involved in the construction of the pan-genome shows that the L9 strain is highly like *Bacillus velezensis* GH1-13 and has significant genetic differences with six other *Bacillus velezensis*. As shown in Figure 8B, the number of common genes increases with the number of genomes, suggesting that *Bacillus velezensis* has an open pan-genome. Figure 8C shows the number of common and endemic ortholog clusters. The center of the flower represents the ortholog cluster, which is present in all strains, and the petals represent those unique to each strain. The results showed that the genomes of *Bacillus velezensis* L9 and seven other strains contained 3158 core genes. During the study, *Bacillus velezensis* L9 was observed to contain 227 specific genes, which were most likely involved in the degradation of ZEN mycotoxins. Specific genes of L9 strains were annotated in the COG database (Figure 8D). In terms of gene function prediction, the largest number of genes was associated with replication, recombination, and repair, followed by defense mechanisms. They are also associated with other gene functions such as transcription, lipid transport, and metastasis. However, the specific function of some genes is not yet clear. This finding confirms the ability of the L9 strain to survive in harsh environments and effectively resist harmful external substances, thus enhancing the adaptability of *Bacillus velezensis* L9 to a wide range of environmental conditions.

## 4. Discussion

ZEN is considered to be one of the most common mycotoxins in feed and food pollutants, which can cause significant harm not only to livestock but also to human health [36]. As a result, detoxification of contaminated fodder and food has become a worldwide trend [37]. In recent years, microbial degradation has become a common choice for feed additives due to its efficiency, low cost, and the fact that it does not produce toxic by-products when degrading mycotoxins. In addition, it can be used to improve the taste of animals and optimize the gut microbiome [38]. Previously, it has been reported that ZEN biodegrading *Bacillus velezensis* A2 was isolated from mold-contaminated soil in Liaoning Province, and that A2 fermentation at 72 h only cleared ZEN in LB medium at 7.45 μg/mL [39]. In their study of fermented soybean products, Chen et al. identified a *Bacillus* sp. B2, which degraded corn contaminants by 58% at a concentration of ZEN of 5 mg/kg [40]. Expression of *Bacillus subtilis* SCK6 peroxidase BsDyP in *E. coli* BL21/p G-Tf2 is capable of breaking down ZEN into the less toxic 15-OH-ZEN [41]. According to previous reports, adding *Bacillus subtilis* ANSB01G to the diet can alleviate ZEN-induced oxidative stress and cell apoptosis in pregnant sows, while reducing fecal ZEN residue [42]. *Bacillus subtilis* ANSB01G can also alleviate the effects of ZEN on sows and mitigate the impact of ZEN on mouse growth performance, serum immune function, antioxidant capacity, and tissue residue [43]. Therefore, the development of new advantageous strains with clear biological characteristics, capable of thriving in harsh environments, holds tremendous significance. In this study, we screened a strain of *Bacillus velezensis* L9 that efficiently degrades ZEN, reducing the solubility by up to 91.14% at 37 °C after 24 h.

In this study, we evaluated the probiotic properties of the L9 strain that efficiently degrades ZEN. The adaptability and genetic stability of bacteria in different culture temperatures, acidities, and bile salt concentrations were compared in terms of bacterial growth and metabolic activity. The study found that the survival of the strain remained at 10^7^ CFU/mL or higher in hot, acidic, and alkaline environments. In the past, zearalenone hydrolase (ZHD) has been reported to mediate the construction of recombinant carboxypeptidase fusions in ASAG of *Bacillus amyloliquefaciens*, an enzyme that completely degrades ZEN in pH 7 at 35 °C [44]. In addition, Cheng et al. cloned the gene ATM from the HNGD-A6 strain and successfully expressed it in BL21, with a degradation rate of 67.82% for ZEN at a pH of 8.5 and 80 °C [45]. This finding confirms that the L9 strain remains highly active in hot, acidic, and alkaline environments, which is a significant advantage among the currently excavated strains.

At present, there are two main methods for the microbial decomposition of ZEN: bacterial adsorption and enzyme degradation. Bacterial adsorption refers to the absorption of specific cell walls by microbes to reduce the toxic content of ZEN. The cell wall contains proteins, lipids, and carbohydrates, which form different adsorption sites and enable hydrogen bonds, ion bonds, and hydrophobic interactions to achieve adsorption. Enzyme degradation mainly occurs when enzymes secreted by bacteria open the ester groups in the ZEN lactone ring and alter the existing structure of ZEN lactone, eliminating its estrogenic properties [46]. To investigate the detoxification of ZEN by *Bacillus velezensis* L9, we identified active components that degrade ZEN. The degradation mechanism of the L9 strain involves enzymatic degradation and adsorption degradation, with enzymatic degradation being the most important. Adsorption degradation showed a higher degradation rate in living cells than in inactivated cells. This difference may be due to the inactivation of the adsorption sites in living cells during heat treatment, leading to a decrease in the ZEN-adsorption capacity. However, the supernatant of the L9 strain showed the best degradation effect, suggesting that the L9 strain’s supernatant plays a key role in the in vitro degradation of ZEN. The fact that high temperatures may reduce microbial activity, and, thus, degradation efficiency, suggests that the main active component degrading ZEN is probably extracellular enzymes. To further confirm the possibility that the active substances are enzymes, we added SDS and protease K to the supernatant by means of chemical and biological degeneration and found that the degradation rate of SDS or protease K decreased in the supernatant alone. When SDS was added with protease K, the degradation rate decreased significantly. Therefore, one or more extracellular enzymes are the main active substances that degrade ZEN. This finding is consistent with previous studies in which *L.paracasei* was able to reduce ZEN by adsorption and convert ZEN into α-ZOL and β-ZOL [47]. The BAMF_RS30125-encoding protein of *Bacillus amyloliquefaciens* H6 belongs to the coenzyme A thiesterase YBGC/FADM family, which is successfully expressed in *E. coli* and degrades into a non-toxic product by breaking ZEN’s lactone bonds or lactone rings [48].

In agricultural production, *Bacillus* is an important class of mycotoxin-degrading strains, and traditional methods of experimental analysis and identification make it difficult to comprehensively analyze Bacillus-resistant substances or fully exploit their antimicrobial potential. Bioinformatics tools make up for these shortcomings, and they can provide an in-depth understanding of the nature and function of organisms from a genomic perspective, bringing new insights into microbial research [49].

The fungal cell wall is a complex polysaccharide structure that plays a crucial role in protecting the survival of fungi, and the disruption of its structure can lead to the rupture of the plasma membrane and cytolysis [50]. Therefore, cell wall-degrading enzymes may serve as important inhibitors for the prevention and control of zearalenone mycotoxins. The main components of fungal cell walls include chitin, β-1,3-glucan, mannan, cellulose, and galactose polymers [51,52], and chitinase hydrolyzes chitin to produce 2-acetylamino-2-deoxy-D-glucose (GlcNAc) [53], and glucanase, cellulase, and peptidases can synergize with chitinase. At the same time, glucoamylase, cellulase, and peptidoglucanase can work synergistically with chitinase to destroy the integrity of the fungal cell wall structure, thus achieving the destruction of zearalenone mycotoxins and having an effect [54]. In this study, 108 genes encoding the CAZy family were predicted from the genome of strain L9, which can encode cellulose synthase (EC 2.4.1.12), chitinase (EC 3.2.1.14), endo-1,3-beta-glucanase (EC 3.2.1.39), beta-glycosidase (EC 3.2.1.21), and peptidoglycan-degradation-related enzymes. Considering the carbohydrate enzymes of Bacillus reported elsewhere, it can be affirmed that these enzymes are beneficial for both animals and humans. Cellulose synthase enzymes have the function of catalyzing cellulose synthesis and secreting polymers through transmembrane channels formed by transmembrane regions [55]. Chitinase not only enhances the ability to resist pathogenic bacteria but also exhibits inhibitory effects on human tumor cells [56]. Beta-glucosidase is a type of cellulase, possessing functions such as antioxidant properties, blood sugar regulation, digestive enhancement, and assisting in alleviating symptoms of anemia [57]. Therefore, the destruction of the cell wall structure by synthesizing cellulase and glucanase may be one of the ways that strain L9 can efficiently degrade zearalenone, while the presence of other cell wall structure-related enzyme genes also suggests that the strain has the potential to degrade the cell walls of other pathogenic fungi.

According to the predicted results of secondary metabolites, L9 can produce 15 different types of secondary metabolites through various pathways, among which most of the confirmed active metabolites with antibacterial activity in the representative *Bacillus velezensis* FZB42 [58] of the same class have been identified. Among the seven highly similar secondary metabolites identified, functional reports have been reported for FZB42 and *Bacillus subtilis* 168. Four of these secondary metabolites have antifungal activities, including three lipopeptides: surfactin, fengycin, bacilysin, and one bacillibactin. In addition, strain L9 could synthesize secondary metabolites with inhibitory effects on bacteria, mainly polyketides, namely, bacillaene, macrolactin, and difficidin. Among them, surfactin is one of the most surfactant-active biosurfactants, known to effectively disrupt fungal biofilms [59]. Macrolactin is a 24-membered macrolide compound that inhibits a wide range of Gram-positive pathogens [60]; bacilysin has broad-spectrum antimicrobial activity against bacteria and fungi, and studies have reported it as the major player in *Bacillus velezensis*-antagonizing Gram-negative foodborne pathogenic bacteria in the antagonism of Gram-negative foodborne pathogens [61]. Bacillaene is a polyene antibiotic that acts by inhibiting protein synthesis and is itself sensitive to light, temperature, and oxygen [62]. Macrolactin is a bacterial peptide deformylase inhibitor, and about 17 different classes have been identified [63], while the more frequently reported is Bacillus Macrolactin A [64], which has been reported more frequently in *Bacillus*, but few studies on Macrolactin H have been described. However, Macrolactin H is rarely studied, and it is initially hypothesized that it may be a new bacteriostatic product, which needs to be confirmed by further studies in the future.

Microorganisms have complex structures and diverse species. Conducting comparative genomics research helps to better understand the differences between microbial species and further explore changes in the evolutionary process of strain genomes, thereby enhancing the ability of strains to adapt to their surrounding environment. Comparative genome analysis shows a high similarity between the L9 strain and *Bacillus velezensis* GH1-13, with significant genetic differences from another six strains of *Bacillus velezensis*. The size of the L9 strain’s genome increases with the increase in the number of genes, indicating that *Bacillus velezensis* L9 undergoes open growth in response to environmental changes. *Bacillus velezensis* L9 contains 227 specific genes, a number higher than the other six strains of *Bacillus velezensis*. These specific genes, identified through gene annotation, are mostly related to replication, recombination, and repair, followed by defense mechanisms. Therefore, comparative genomics analysis further validates the adaptation of *Bacillus velezensis* L9 to a wide range of environmental conditions.

Overall, after 24 h of culture at 37 °C, *Bacillus velezensis* L9 was 91.14% more efficient at degrading ZEN (10 μg/mL), and the strain survived in high-temperature, acidic, and choline environments. This strain can produce multiple enzymes and antibacterial secondary metabolites, which enables it to efficiently remove ZEN in terms of both adsorption and enzymatic degradation. By comparing genomics, we further identified the L9 strain as *Bacillus velezensis* and gained a clear understanding of its genetic background. Therefore, adding this strain to ZEN-contaminated feed as a probiotic could provide some technical support for the control of mycotoxin contamination in food grains.

## 5. Conclusions

The *Bacillus velezensis* L9 selected in this study is a novel ZEN-degrading bacterium whose extracellular enzymes are key to the degradation of ZEN. It degrades 91.14% of ZEN within 24 h. It was found that the genome of the L9 strain contains many secondary metabolite synthesis gene clusters and numerous other hydrolase-coding genes, providing a valuable basis for further explanation of the bacteriostatic ability of the L9 strain and its ability to produce many kinds of hydrolases. In addition, comparative genomics analysis revealed a high similarity between the L9 strain and *Bacillus velezensis* GH1-13, suggesting that the bacteria remained highly transgenic over a long evolutionary period, giving us a better understanding of the genetic background of the strain. This study provides a solid basis for the further development and utilization of the L9 strain to degrade maize gibberellin.

## Figures and Tables

**Figure 1 foods-13-00530-f001:**
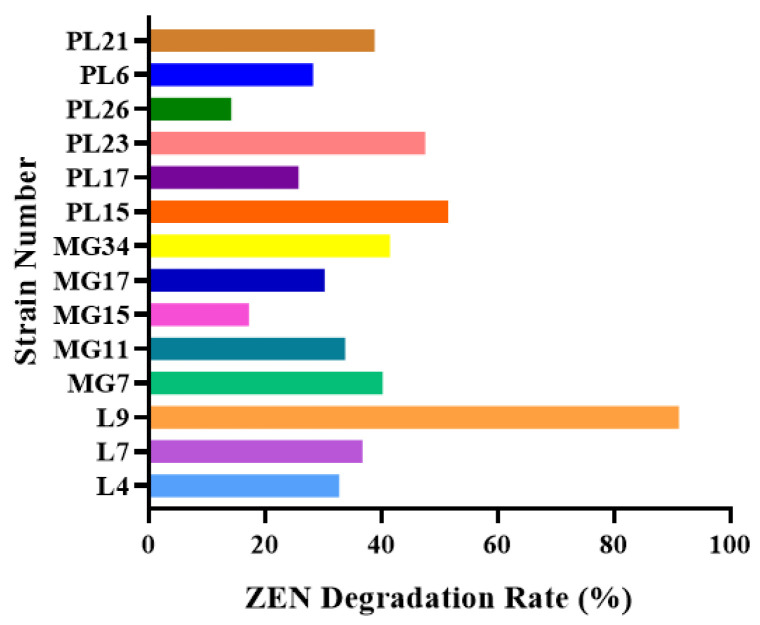
Degradation rate of ZEN by isolated strains.

**Figure 2 foods-13-00530-f002:**
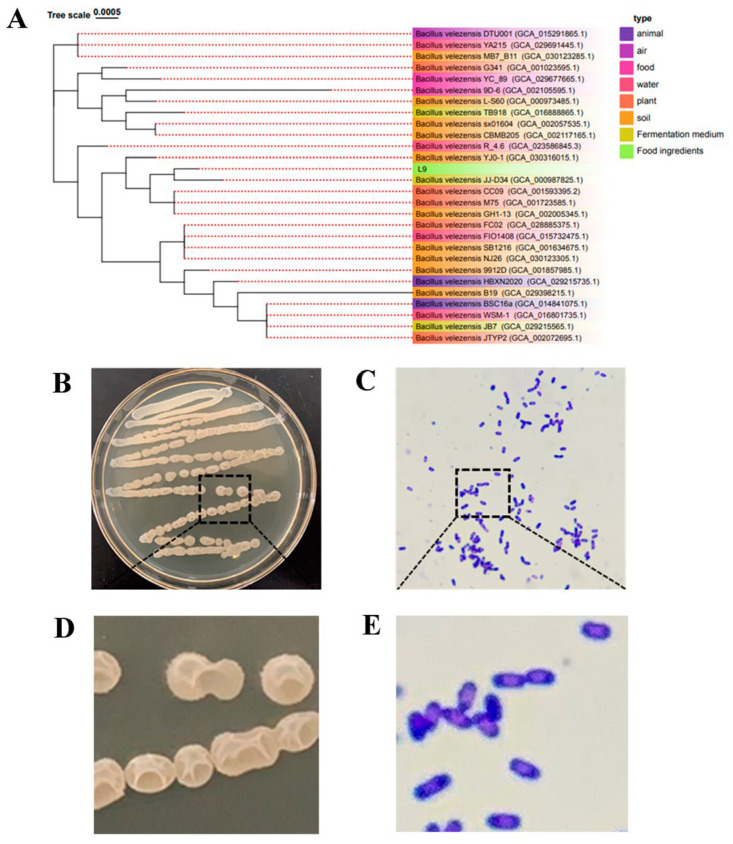
The phylogenetic tree and morphological characteristics of L9 strain. (**A**) Phylogenetic tree of the based on 16S rRNA gene sequences. (**B**,**D**) Cultures of L9 strain grown on LB agar medium for 24 h at 37 °C. (**C**,**E**) Gram staining results of L9 strain under a light microscope (100×, 1000×).

**Figure 3 foods-13-00530-f003:**
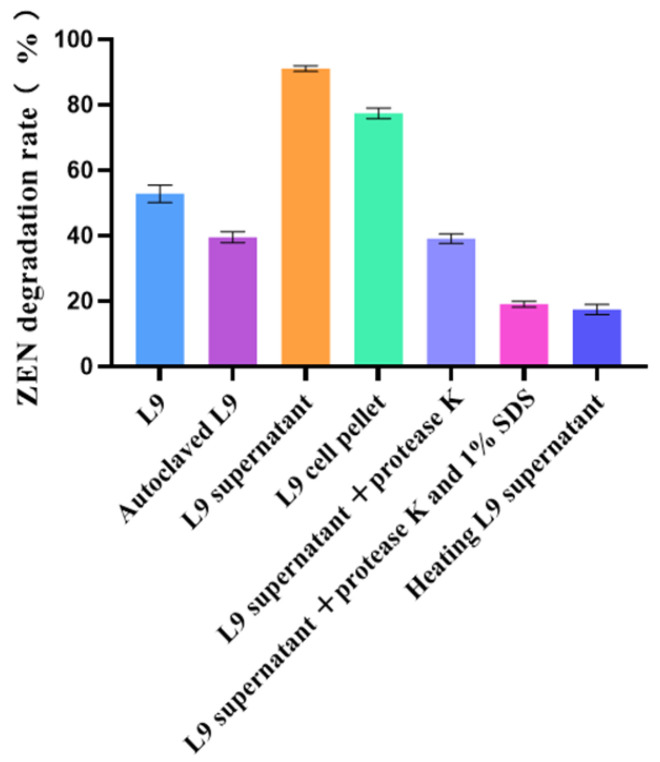
Effects of different treatments on ZEN detoxification by L9 strain.

**Figure 4 foods-13-00530-f004:**
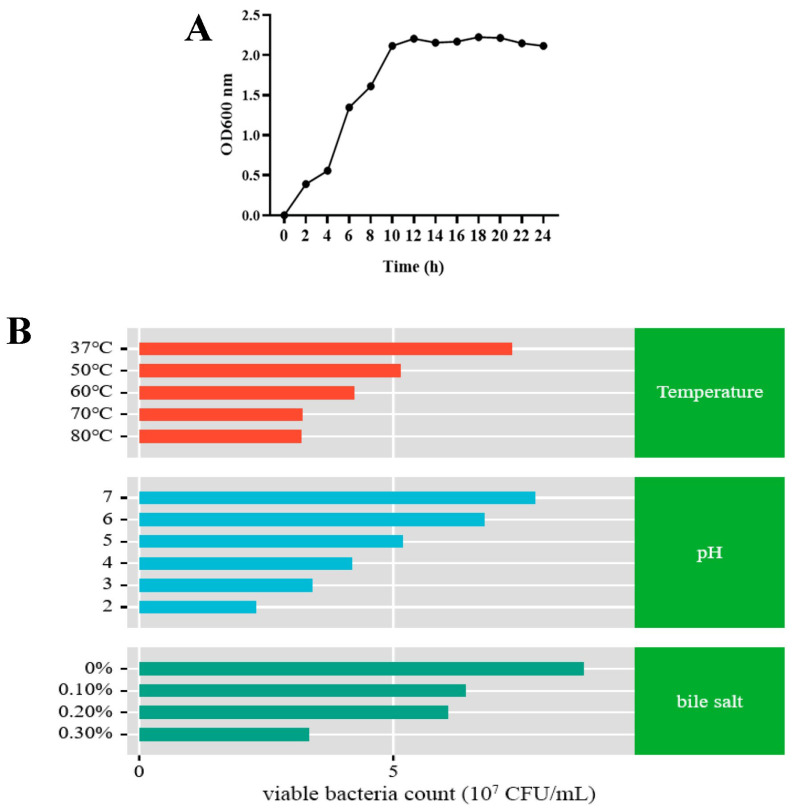
Biological characteristics of L9 strain. (**A**) Growth curve of L9 strain. (**B**) Tolerance assay of L9 strain. Tests for resistance to temperature, acid, and bile salts were conducted by assessing the OD600 value using a spectrophotometer and incorporating it into the regression formula to determine the count of living bacteria.

**Figure 5 foods-13-00530-f005:**
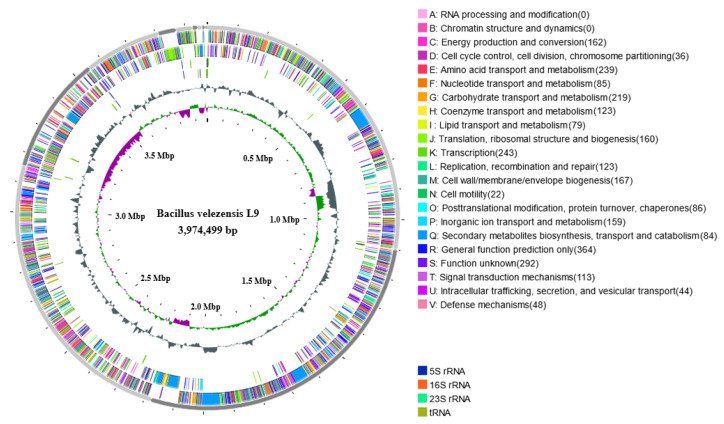
The circle graph of *Bacillus velezensis* L9 genome chromosome. The genome chromosome loop consists of seven circles, from the outside to the inside. The first circle represents the COG-diluted genes in the justice chain. The second circle represents the COG annotated genes in the antisense chain. The third circle represents the non-coding RNA in the justice chain. The fourth circle represents the non-coding RNA on the antisense chain. The fifth circle represents GC content. The sixth circle represents GC-SKEW, green is positive, and purple is negative. The seventh circle represents the length of the chromosome sequence.

**Figure 6 foods-13-00530-f006:**
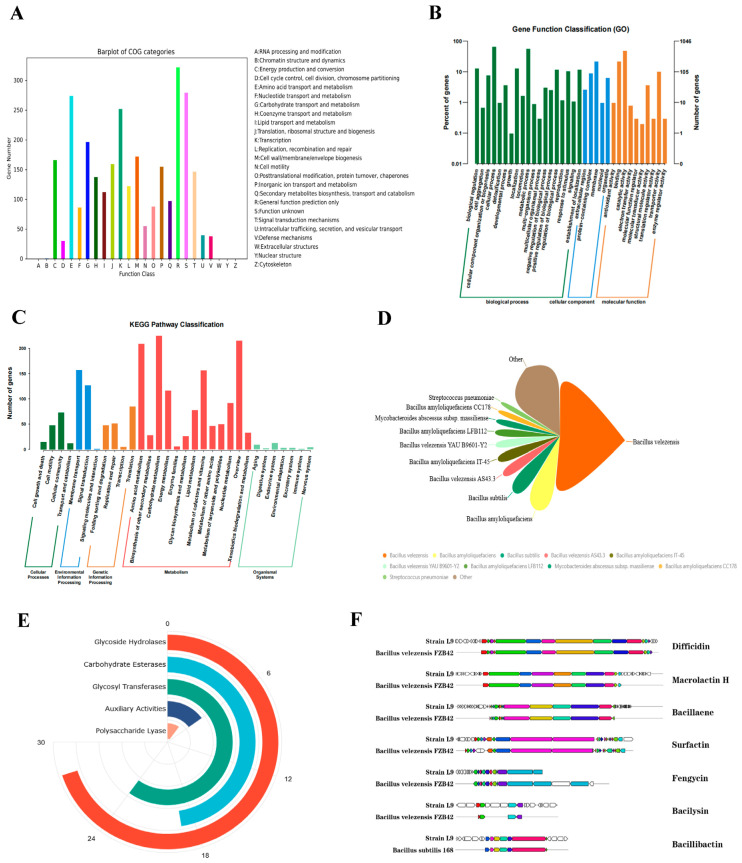
Genome annotation of L9 strain. (**A**) COG function classification of protein-coding genes. (**B**) GO function classification of protein-coding genes. (**C**) KEGG pathway classification of protein-coding genes. (**D**) Pie chart of the homology distribution. (**E**) CAZy classification result. (**F**) Predictive analysis of secondary metabolite gene clusters.

**Figure 7 foods-13-00530-f007:**
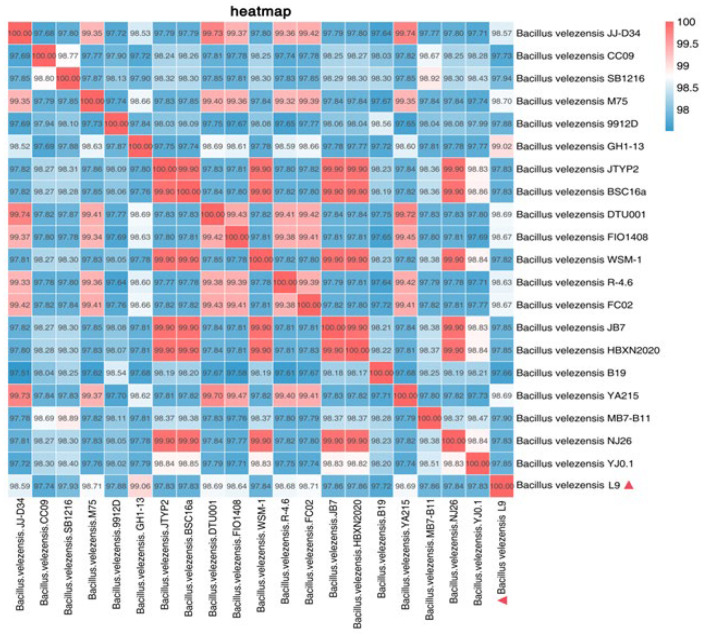
ANI values (%) between L9 strain and other 20 similar strains.

**Figure 8 foods-13-00530-f008:**
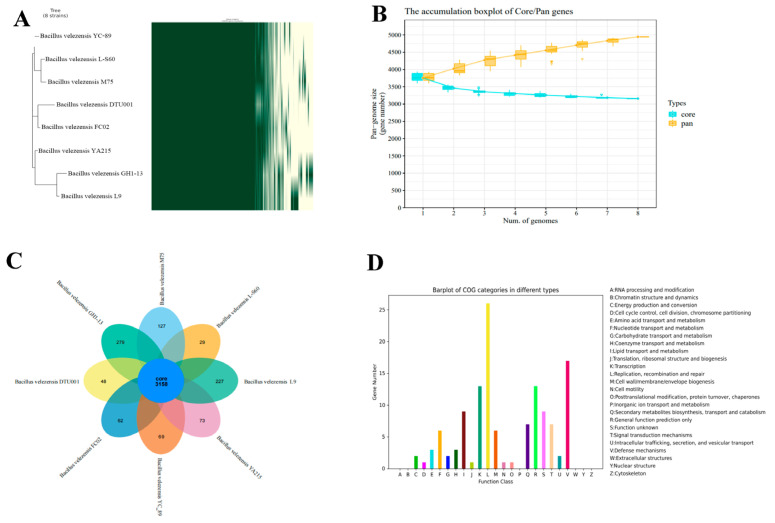
The pan-genomes of eight *Bacillus velezensis* were determined using the Roary matrix. (**A**) A heatmap showing the gene presence (green) in each of the 8 strains. A phylogeny built based on the core genes is shown on the left. (**B**) Core-pan gene dilution box chart. (**C**) Number of ortholog clusters common and specific to each of the 8 strains. Flower centers indicate ortholog clusters that are common to all strains, and petals indicate those that are unique to each strain. (**D**) COG annotation of genes specific to the L9 strain.

**Table 1 foods-13-00530-t001:** The general features of the *Bacillus velezensis* L9 genome.

Feature	Chromosome
Genome size (bp)	3,974,499
G+C content (%)	46.25
Protein-coding genes (CDS)	4082
Total gene length (bp)	3,561,637
Gene/genome (%)	89.61
5S rRNA	8
16S rRNA	1
23S rRNA	1
tRNA	81

**Table 2 foods-13-00530-t002:** Identification of the region of secondary metabolite synthesis in L9 strain.

Cluster ID	Genomic Site		Product Type	Most Similar Known Gene Cluster		
	Initial Site	Terminal Site		Most Similar Product	Similarity, %	Source
Cluster 1	382,141	427,027	NRPS	—	—	—
Cluster 2	573,274	667,047	transAT-PKS	difficidin	100	*Bacillus velezensis* FZB42
Cluster 3	928,854	969,954	T3PKS	—	—	—
Cluster 4	1,033,593	1,055,476	terpene	—	—	—
Cluster 5	1,078,046	1,101,559	NRPS	plipastatin	38	—
Cluster 6	64,361	105,605	PKS-like	butirosin A/ butirosin B	7	—
Cluster 7	187,671	208,411	terpene	—	—	—
Cluster 8	504,977	591,316	transAT-PKS	macrolactin H	100	*Bacillus velezensis* FZB42
Cluster 9	814,801	915,529	transAT-PKS	bacillaene	100	*Bacillus velezensis* FZB42
Cluster 10	980,011	1,078,369	NRPS	fengycin	86	*Bacillus velezensis* FZB42
Cluster 11	298,436	339,854	other	bacilysin	100	*Bacillus subtilis* 168
Cluster 12	859,024	910,818	NRPS	bacillibactin	100	*Bacillus velezensis* FZB42
Cluster 13	194,589	259,996	NRPS	surfactin	82	*Bacillus velezensis* FZB42
Cluster 14	70,747	111,868	ladderane	—	—	*Bacillus velezensis* FZB42
Cluster 15	1	14,773	NRPS	fengycin	20	—

## Data Availability

The data supporting the findings of this study are available within the article. The 16S rRNA gene sequence of the strain was deposited into the GenBank database. The high-quality raw data genome sequence for *Bacillus velezensis* L9 was deposited into NCBI under accession GCF_033022835.1.

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
