# Peer review of "A Novel Bacillus Velezensis for Efficient Degradation of Zearalenone"

_foods, 2024, doi:10.3390/foods13040530_

Round 1

Reviewer 1 Report

Comments and Suggestions for Authors

The manuscript describes the zearalenone degradation capacity of Bacillus velezensis and the characterization of its genome. The results show high percentages of mycotoxin reduction and the presence of genes that may be directly related to its inactivation. The contribution of this work is observed in the evaluation of an alternative for the management of grains and raw materials used in the production of food and rations.

The work is well planned, with an adequate experimental design and using high-resolution molecular characterization methodologies.

In relation to the methods used, it is mentioned that the main criterion for the isolation of the strain and the evaluation of its degradation capacity was the residual analysis of Zearalenone, using HPLC with a fluorescence detector. The above is adequate, however, information on possible degradation metabolites cannot be obtained. Likewise, the characterization of the metabolites would allow a better correlation with the gene families and activities of the enzymes found in the genome.

It is well known that the main metabolites of zearalenone are alpha and beta zearalenol, which have a much higher estrogenic capacity, even than zearalenone itself. Given that there are no results on the metabolites produced, the potential use of the bacteria as a probiotic is limited. In this sense, I suggest that with the information available in the fluorescence chromatograms, the possibility of production of said metabolites be ruled out.

A thorough review of punctuation throughout the manuscript is suggested.

Additionally, the following suggestions are made

Page 1, Line 29. Pull apart “decades[1]”, do the same throughout the manuscript.

Page 1, Line 37. Change is to are

Page 1, Line 40. You refer to zinc or ZEN?. This information is not supported by any reference.

Page 1, Line 45. Correct TIf

Page 2, Lines 90-92. It is suggested to describe the method of extraction of ZEN from agar and indicate the reference of the technique used for the quantification of zearalenone..

Page 3, Line 121. It is suggested to describe the characteristics of the centrifuge and the rotor used..

Page 3, Line 146. Pull apart “Cand”.

Page 4, Line 149. Pull apart “600nm”.

Page 4, Line 168. One point missing after process.

Page 5, Line 215. Pull apart “16SrRNA”

Page 6, Line 147. Figure foot and Figure are in disarray.

Page 6, Figure 1. Activity is described as degradation rate and results are given in percentage units. I suggest that the description be consistent.

Page 8, Line 319. Temperature y-axis values do not appear complete.

Page 8, Line 329. Pull apart “Figure.5”.

Page 10, Line 413. Correct spelling.

Page 11, Line 463. Pull apart “database(Figure 8D)”.

Reviewer 2 Report

Comments and Suggestions for Authors

Dear authors

The paper is fine and informative, meanwhile for any expert, the selection process seams not correct.

You have used LB medium which is a general medium, where most bacteria can grow on it.

You must argument in deep why you have used the LB medium as a supportive medium for bacterial growth. This part must be well written and well explained.

The other parts of the paper are fine and systematic.

No need for English editing.

You need to rewrite nearly all the bacterial species’ names as italic.

With my pleasure

Reviewer 3 Report

Comments and Suggestions for Authors

The manuscript addressed  a Novel Bacillus velezensis that  demonstrated high Zearalenone degradation efficiency. The manuscript is generally good. However, the following points need to be resolved by the authors

1- The abstract should be revised highlighting the aim of the study.

Briefly mention the isolation process and the rational of selecting this particular strain. 

 -2Elaborating on the biodegradation process and the role of identified enzymes would be informative.

--Introduction Consider reorganizing the information for better flow. Start with the global context, then dive into specific details about ZEN.

consider briefly mentioning its economic and social impact or potential solutions like preventative measures or detoxification methods.

Discussion should be enriched by addressing the specific health risks associated with ZEN.

Provide an example that actively pursuing detoxification.

Clarify the type of harsh environments relevant for  potential.

The scientific names throughout the entire manuscript need to be all italic 

Comments on the Quality of English Language

Minor editing of English language required

Round 2

Reviewer 1 Report

Comments and Suggestions for Authors

The authors made the recommended changes and adjustments.